# Synthetic Calcium Phosphate Ceramics as a Potential Treatment for Bisphosphonate-Related Osteonecrosis of the Jaw

**DOI:** 10.3390/ma12111840

**Published:** 2019-06-06

**Authors:** Siri Paulo, Mafalda Laranjo, Ana M. Abrantes, João Casalta-Lopes, Kathleen Santos, Ana C. Gonçalves, Anabela Baptista Paula, Carlos Miguel Marto, Ana Bela Sarmento-Ribeiro, Eunice Carrilho, Arménio Serra, Maria F. Botelho, Manuel M. Ferreira

**Affiliations:** 1Institute of Endodontics, Institute for Clinical and Biomedical Research (iCBR), Area of Environment Genetics and Oncobiology (CIMAGO), CNC.IBILI, Faculty of Medicine, University of Coimbra, 3000-548 Coimbra, Portugal; m.mferreira@netcabo.pt; 2Biophysics Institute, Institute for Clinical and Biomedical Research (iCBR), Area of Environment Genetics and Oncobiology (CIMAGO), CNC.IBILI Consortium, Faculty of Medicine, University of Coimbra, 3000-548 Coimbra, Portugal; mafaldalaranjo@gmail.com (M.L.); mabrantes@fmed.uc.pt (A.M.A.); joao.casalta@gmail.com (J.C.-L.); kathleen.santos@hotmail.com (K.S.); mfbotelho@fmed.uc.pt (M.F.B.); 3Radiation Oncology Department, Coimbra University Hospital Center, 3000-548 Coimbra, Portugal; 4Applied Molecular Biology Unit, Institute for Clinical and Biomedical Research (iCBR), Area of Environment Genetics and Oncobiology (CIMAGO), CNC.IBILI Consortium, Faculty of Medicine, University of Coimbra, 3000-548 Coimbra, Portugal; acc.goncalves@gmail.com (A.C.G.); absarmento@fmed.uc.pt (A.B.S.-R.); 5Institute of Integrated Clinical Practice, Institute for Clinical and Biomedical Research (iCBR), Area of Environment Genetics and Oncobiology (CIMAGO), CNC.IBILI, Faculty of Medicine, University of Coimbra, 3000-548 Coimbra, Portugal; anabelabppaula@sapo.pt (A.B.P.); eunicecarrilho@gmail.com (E.C.); 6Institute Experimental Pathology, Institute for Clinical and Biomedical Research (iCBR), Area of Environment Genetics and Oncobiology (CIMAGO), CNC.IBILI, Faculty of Medicine, University of Coimbra, 3000-548 Coimbra, Portugal; mig-marto@hotmail.com; 7Department of Chemical Engineering, Faculty of Science and Technology University of Coimbra, Polo II, Pinhal de Marrocos, 3030-790 Coimbra, Portugal; armenio.serra@gmail.com

**Keywords:** biomaterials, biomineralization, bisphosphonate-related osteonecrosis of the jaw, cellular biology, gingival fibroblasts, osteonecrosis, zoledronate

## Abstract

(1) Background: Bisphosphonate-related osteonecrosis of the jaw (BRONJ) is one of the most often seen side effects in patients treated with nitrogen-containing bisphosphonates (BPs), a post-surgical non-healing wound condition. Since calcium phosphate (CP) compounds are able to adsorb zoledronate (ZOL) when used as a drug delivery vehicle, we aimed to verify if these ceramics might have a potential protective effect for soft tissues surrounding surgical osseous wounds. (2) Methods: The chemical reaction between ZOL and CP compounds was evaluated through ultraviolet-visible spectroscopy and elemental analysis. A primary culture of human gingival fibroblasts (HGF) was established as a model to evaluate the cytotoxicity of the association of ZOL (5–500 μM) and of ZOL/biphasic calcium phosphates (BCP). Metabolic activity, cell viability, types of cell death, the cell cycle through, and the migration ability of human gingival fibroblasts were evaluated. (3) Results: ZOL was adsorbed by biphasic calcium phosphate compounds in an aqueous solution. The HGF were sensitive to ZOL toxicity; nevertheless, ZOL/BCP showed a significant protective effect regarding metabolic activity, cell viability, and cell migration. (4) Conclusions: BCP interaction with ZOL reduces or abolishes its toxicity in HGF. This finding represents a potential solution for BRONJ in the case of patients undergoing therapy with ZOL.

## 1. Introduction

The use of bisphosphonates (BPs) has dramatically increased over the past few years. BPs offer substantial clinical benefit when an imbalance between osteoblast-mediated bone deposition and osteoclast-mediated bone resorption underlies physiopathology [1]. The main impact of this therapy is the prevention and treatment of cancer-related skeletal complications associated with bone metastases such as pathological fractures, spinal cord compression, tumour induced hypercalcaemia, and severe bone pain [2].

Biochemical interactions between bone microenvironment and cancer cells promote bone destruction and tumour growth [3]. In patients with advanced malignancy, tumour cells invade the bone, yielding skeletal metastases and disrupting bone homeostasis. Osteoclasts and osteoblasts are recruited, leading to the liberation of growth factors and cytokines which can feed back to metastasis growth [4]. The pharmacological effect of BPs is related to their binding to the inorganic component of bone and to their biochemical effect on cells, predominantly osteoclasts [5]. When intravenous BPs are administered, the reduction of activity and number of osteoclasts decreases bone reabsorption, diminishing the malignancy and inhibiting the growth of bone metastasis [1,6].

The propensity of the jaws to bisphosphonate-related osteonecrosis of the jaw (BRONJ) may be due to several anatomical and physiological factors. BPs tend to be highly concentrated in the jaws rather than other skeletal sites because they preferentially deposit in bones with high turnover rates and in sites of significant remodelling [7,8].

Thus, the forces of masticatory function require a rapid bone turnover and can easily induce microfractures that also need remodelling [9,10]. Furthermore, unlike other skeletal sites, after surgery or trauma, the wound can be continually exposed to more than 500 different species of microorganisms, resulting in a high susceptibility to contamination and infection [11]. In normal circumstances, the healing of an open bony wound in the presence of normal oral microflora occurs without complication, although, in these cases, when oral microbes reach necrotic bone, they hinder healing [7,8]. The prevalence of BRONJ in cancer patients receiving intravenous zoledronate following tooth extraction ranges from 1.6% to 14.8% [12]. Currently there is no effective treatment for BRONJ. The approaches available are mainly conservative (mouth rinses and antibiotics), surgical interventions, and adjuvant strategies, such as hyperbaric oxygen therapy [13].

BRONJ, commonly following dental surgery, has been reported in patients receiving intravenous BPs, particularly zoledronic acid (ZOL), even after the end of the therapy. BPs released from oral bone may impair the migration and growth of oral fibroblasts, inhibiting matrix deposition and the remodelling of oral soft tissues, a critical aspect of wound healing [14]. For the patients, these translate into signs and symptoms such as pain, bone exposure, gingival swelling, soft tissue ulceration, fistular trajectories, tooth mobility, paresthesia, and anesthesia [15].

Presently, there is insufficient evidence to support a particular regimen for the treatment of BRONJ [16]. BP–calcium phosphate (CP) composites are used in the treatment of osteoporosis. The affinity of BPs for hydroxyapatite (HA) explains the basis of this therapy that is used in the treatment of such bone diseases [17]. CP drug delivery systems ensure the controlled release of BPs, which are used due to their anti-resorptive properties [18]. CP ceramics are good delivery systems for drugs and growth factors because of their biocompatibility, variable stoichiometry, surface charge density, functionality, and dissolution properties [17,19].

Considering the above, we hypothesize that CP ceramics might be used in the treatment of BRONJ. The chemical binding of BPs to CP ceramics might result in the decrease of BP availability in the surgical wound and, therefore, in a lower toxicity. To evaluate our hypothesis in vitro, we established primary cultures of human gingival fibroblasts (HGF) that were used as a model to evaluate BP toxicity and the potential protective effect of biphasic calcium phosphates. 

## 2. Materials and Methods

To establish primary cultures of HGF, the biological material was obtained from human gingival biopsies. The protocol for obtaining such biopsies was approved by the Ethics Committee of the Faculty of Medicine, University of Coimbra (Of Ref. 73-CE-2012). Patients selected for these procedures had indications of gingivectomy and fulfilled the following criteria: Good oral hygiene, non-smokers, and without any known systemic disorders. Before the surgery, patients were informed about the project and chose to participate of their own free will, having read, understood, and signed the respective informed consent.

### 2.1. Materials

Zoledronic acid [(1-hydroxy-2-imidazol-1-yl-1-phosphonoethyl)phosphonic acid] were obtained from Hangzhou Pharm & Chem Co., Ltd., (Hangzhou, China). Two calcium phosphate ceramics were used; one was constituted of 99.9% beta-tricalcium phosphate (TCP; Adbone^®^TCP, Medbone^®^, Medical Devices; Leiria, Portugal), and the other was constituted of 75% hydroxyapatite (HA) and 25% biphasic calcium phosphates (BCP; Adbone^®^BCP, Medbone^®^, Medical Devices, Leiria, Portugal).

### 2.2. ZOL-Ceramics Adsorption

The reaction of ZOL with calcium phosphate ceramics was studied. Aqueous solutions of ZOL in a concentration ranging from 0.1 to 1 mM were prepared. To each 5 mL of ZOL solutions, 0.02 g of the TCP, 0.02 g of the BCP, or 0.06 g of the BCP was added, and the mixtures were stirred at room temperature overnight.

Visible/UV spectroscopy (Hitachi U200 spectrophotometer, Chiyoda-ku, Tokyo, Japan) studies were performed to measure ZOL concentration, allowing an evaluation of the incorporation of ZOL in the ceramics. In order to confirm the results obtained, an elemental analysis was performed (Fisons O.CHS Element Analyzer, Thermo Fisher Scientific, Waltham, MA, USA).

### 2.3. Primary Cell Cultures

Primary cultures of HGF were established through adaptation of previous works [20,21]. The biopsies were transported to the laboratory in phosphate-buffered saline (PBS; in mM: 137 NaCl, 2.7 KCl, 10 Na_2_HPO_4_, and 1.8 KH_2_PO_4_ [pH 7.4]), supplemented with 1% penicillin/streptomycin/amphotericin B (A-5955, Sigma Aldrich^®^, Israel).

The explants were scraped to remove any epithelial tissue, cut into pieces, and submitted to digestion by incubation with 2% collagenase type I (Affymetrix^®^ 13820, Cleaveland, OH, USA) at 37 °C for 45 min. Subsequently, collagenase was carefully replaced by a solution of 0.25% trypsin (Gibco^®^ 25200056, Thermo Fisher Scientific, Paisley, Scotland), and digestion continued for 15 min. Afterwards, the fragments were distributed through a petri dish, and a few drops of DMEM (Dulbecco’s Modified Eagle’s Medium, D-5648, Sigma Aldrich^®^, Saint Louis, MO, USA), supplemented with 10% fetal bovine serum (FBS, Sigma Aldrich^®^, F7524, Brazil), and 1% penicillin/streptomycin/amphotericin B were added. The cultures were maintained at 37 °C in a humidified incubator with 5% CO_2_ and filled with a medium after 24 h. Cultures were monitored, and the medium was replaced every three days. Seventh to eighth passages HGF were used in this study. 

### 2.4. ZOL and ZOL–BCP Treatment

The primary cell cultures were plated in 48-well plates at a density of 2 × 10^4^ cells per well and incubated overnight to allow cell attachment. ZOL was administered at concentrations of 5, 25, 50, 100, and 500 µM by dilution in the culture media. In order to evaluate the potential protective effect of biphasic phosphate ceramics (BCP), primary cell cultures were submitted to the ZOL–BCP solutions prepared as described in the ZOL-ceramics adsorption section. Cells were then incubated for periods of 24, 48, 72, 96, and 120 h and evaluated according to the studies subsequently described (Cell Citotoxicity, Cell migration studies/Scratch assay, Viability assay, Cell cycle analysis, Types of cell death).

### 2.5. Cell Citotoxicity

The metabolic activity was determined by MTT assay (3-(4,5-dimethylthiazol-2-yl)-2,5-diphenyl tetrazolium bromide; MTT, Sigma Aldrich^®^, USA) [22,23,24]. Briefly, cell cultures were washed with PBS and incubated with 0.5 mg·mL^−1^ MTT, pH = 7.4, at 37 °C for 4 h. Formazan crystals were solubilized in 40 mM hydrochloric acid in isopropanol. Absorbance was evaluated in a Biotek^®^ Synergy HT microplate reader (Biotek Instruments, Winoosky, VT, USA) at 540 nm with a reference filter of 690 nm. The results allowed us to establish dose-response curves and to calculate IC_50_ values, the concentration required to inhibit cell proliferation by 50%.

### 2.6. Cell Migration Studies/Scratch Assay

Cells were plated in 6-well plates at a density of 1 × 10^6^ cells per well and incubated overnight to allow cell attachment, as described by Liang et al. [25]. The center of the confluent cell monolayers was scratched, and cell cultures were treated as described. Selected areas were marked and photographed immediately (0 h) and, subsequently, every 24 h to record cellular migration into the wounded areas until 120 h, using an ECLIPSE TS100 Nikon microscope (Nikon, Tokyo, Japan) and dedicated software. Cell migration (space between scratch edges) was quantified by measuring the distance between both leading edges using ImageJ software (National Institutes of Health) [26].

### 2.7. Viability Assay

For the evaluation of total protein content, a sulphorhodamine B assay (SRB) was performed [27,28]. Cell cultures were washed with PBS and then fixed with 1% acetic acid in methanol for 30 min at 4 °C. Afterwards, 0.4% SRB (Sigma Aldrich^®^, USA), dissolved in 1% acetic acid, was added and maintained for 1 h in the dark, at room temperature. Then, the cells were washed in tap water, and the SRB crystals were dissolved in 10 mM Tris-NaOH, pH 10. Absorbance was evaluated in a Biotek^®^ Synergy HT microplate reader at 540 nm.

### 2.8. Cell Cycle Analysis

For the analysis of DNA content, 2 × 10^6^ cells were collected, centrifuged, and fixed with ice-cold 70% ethanol for 30 min in the dark. Afterwards, the cells were washed with PBS and incubated with a PI/RNase solution (Immunostep, Salamanca, Spain) for 15 min at room temperature. Cells were analyzed in the FACSCalibur cytometer (BD Biosciences, San Jose, CA, USA) with an excitation wavelength of 488 nm and an emission filter of 585/42. Untreated cell cultures and cells treated with 100 and 500 µM of ZOL or ZOL/BCP for 96 and 120 h were analyzed [29,30].

### 2.9. Types of Cell Death

For the analysis of cell viability and types of cell death, 2 × 10^6^ cells were collected, centrifuged, and washed with PBS prior to incubation with the binding buffer, annexin-V and propidium iodide, (Immunostep, Salamanca, Spain) for 15 min, at 4 °C in the dark. 400 mL PBS were added, the samples were analyzed in the flow cytometer using an excitation wavelength of 488 nm, and emission filters of 530/30 nm and 585/42 nm were used for annexin-V and for propidium iodide, respectively [31,32]. Untreated cell cultures and cells treated with 100 and 500 µM of ZOL or ZOL/BCP for 96 and 120 h were analyzed.

### 2.10. Statistical Analysis

A statistical analysis was performed using the software IBM SPSS (Version 20, IBM Corporation, New York, NY, USA). Values are expressed using measures of central tendency and dispersion. The normality of distributions was evaluated by the Shapiro–Wilk test. The metabolic activity results were compared with the Student t test for an average. IC50 values were compared using the one-factor ANOVA test. A comparison of ZOL and its respective ZOL/BCP metabolic activity and viability values was performed with the Student t test for independent samples if there was there normality of distribution and homogeneity of variances; otherwise, it was performed by the Mann–Whitney test. Regarding the scratch assay, both a types of cell death and cell cycle comparison of the experimental conditions were performed with a one-factor ANOVA test (nonparametric test) or the Kruskal–Wallis test (nonparametric test). All multiple comparisons were corrected according to Bonferroni. A 5% significance level was considered for all comparisons.

## 3. Results

### 3.1. ZOL-Ceramics Adsorption

The absorbance of the ZOL solutions remained constant up to 120 h. Furthermore, different light exposures and temperatures did not influence these results (data not shown). The presence of phosphate ceramics led to the alteration of the absorbance spectra of the ZOL solutions, as can be observed in Figure 1.

Regarding the mixture with TCP, the absorbance decreased 43% after one hour of stirring relative to the ZOL solution, and it gradually decreased up to 87% after 120 h. BCP also led to the decrease in absorbance of the ZOL solutions, with the mixture of 0.06 g being superior, where a fall of 92% after 120 h was observed compared to the mixture of 0.02 g, where a decrease of 60% occurred.

An elemental analysis of BCP samples submitted to ZOL showed the presence of organic atoms, namely nitrogen at a percentage of 0.31 and carbon at a percentage 0.66 (mean values of four trials).

### 3.2. Primary Cell Cultures

The biopsy fragments gave rise to cells with fibroblast morphology, growing attached to the substrate, bipolar, or multipolar and with elongated shapes. About four weeks after the establishment of the cultures, the cells formed a confluent layer exhibiting a spindle form with a central nucleus. Different phases of the primary culture and cell morphology can be observed in Figure 2.

### 3.3. ZOL Citotoxicity

Dose-response curves presented in Figure 3 show the cytotoxicity of ZOL after 24, 48, 72, 96, and 120 h of incubation. 

ZOL cytotoxicity is concentration and time dependent. For the incubation times of 24 and 48 h, it was not possible to calculate the median inhibitory concentration of the cultures (IC50), which was higher than the maximum value tested. However, after 72, 96, and 120 h of incubation, it was possible to calculate the corresponding IC50 values, which are shown in Table 1, with the respective confidence intervals at 95%. The IC50 value obtained for 72 h is significantly higher than the values obtained for 96 (*p* = 0.016) and for 120 h (*p* = 0.013), with no differences between these times.

### 3.4. ZOL–BCP Citotoxicity

Figure 4 shows the comparison of the metabolic activities obtained for the HGF submitted to the treatment with ZOL and with ZOL/BCP (0.02 g) for 48, 72, 96, and 120 h. BCP showed a significant protective effect from ZOL.

After 72 h of incubation, a metabolic activity of 69.3 ± 11.8% was obtained after incubation with 100 μM of ZOL; with the association of BCP, a value of 109.2 ± 11.8% (*p* < 0.001) was obtained. Regarding the administration of 500 μM of ZOL, cellular metabolic activity reached a value of only of 26.7 ± 8.02%; however, the corresponding association with BCP maintained a value of 113.2 ± 13.1% (*p* < 0.001). After 96 h of ZOL incubation, the protective effect of BCP was observed for the concentrations ranging from 25 to 500 μM. This effect was maintained even at 120 h of incubation for all the concentrations of ZOL studied.

### 3.5. Cell Migration Studies/Scratch Assay

Figure 5 shows the percentage of migration of HGF cultures under the treatment with ZOL versus with ZOL/BCP. Untreated cell cultures can recover confluency in 120 h.

However, in cultures subjected to ZOL, confluence recovery is compromised proportionally to concentration and is significant for higher concentrations. Thus, for the concentration of 100 μM, there was a lower percentage of HGF migration after 72 h (*p* = 0.004), 96 h (*p* < 0.001), and 120 h (*p* = 0.003), and, for the concentration of 500 μM, the percentage was lower for the same time points (*p* < 0.001, *p* < 0.001 and *p* = 0, 016, respectively). After treatment with the lower concentrations of 50 μM and 25 μM, the same effect was observed after 96 h (*p* < 0.001) and 120 h (*p* < 0016 and *p* = 0.037, respectively).

Nevertheless, cultures submitted to ZOL/BCP did not show any differences in HGF migration regarding untreated cell cultures. Furthermore, even cultures treated with ZOL100/BCP and ZOL500/BCP for 72 h showed a significantly superior closure area than those treated with ZOL at the same concentration (*p* < 0.001 and *p* = 0.006, respectively). Moreover, the area covered by the cells treated with ZOL (100 μM)/BCP is significantly higher than the cell cultures treated with ZOL alone (*p* < 0.001) after 96 h of incubation.

### 3.6. Viability

Figure 6 shows HGF viability after treatment with ZOL and with ZOL/BCP (0.02 g). After 48 h of incubation, the viability of HGF treated with ZOL/BCP was significantly higher than the cultures incubated with ZOL in the concentrations of 5 μM (*p* = 0.003), 25 μM (*p* < 0.001), 50 μM (*p* < 0.001), and 100 μM (*p* < 0.001), respectively. Regarding the incubation time of 72 h, the viability of HGF treated with ZOL/BCP was superior to the cultures treated with ZOL in the concentrations of 50 μM (*p* < 0.001), 100 μM (*p* < 0.001), and 500 μM (*p* < 0.001). In addition, after 96 h and 72 h of incubation in the concentrations of ZOL where a decrease of viability was observed, there was also a significant protective effect of ZOL/BCP. This was observed after 72 h even with 500 μM, where ZOL led to a viability of only 38.9 ± 13.2%, and the incubation with ZOL/BCP displayed a value of 133.9 ± 23.4% (*p* < 0.001). After 120 h of ZOL incubation, the inhibitory effect led to viabilities of 26.6%, 18.7%, 19.7%, 21.6%, and 14.3% for the concentrations of 5 μM, 25 μM, 50 μM, 100 μM, and 500 μM, respectively, while the ZOL/BCP association led to 96.7% (*p* < 0.001), 108.0% (*p* < 0.001), 128.5% (*p* < 0.001), 95.6% (*p* < 0.001), and 61.3% (*p* = 0.002).

### 3.7. Cell Cycle

As can be seen in Figure 7, cells incubated with the higher ZOL concentration (500 μM) were partially retained in the G0/G1 phase, with a population of 85.7 ± 3.5% versus 66 ± 12.5% (*p* = 0.048) in the control. The same occurred after 120 h of incubation with a population of cells in the G0/G1 phase of 73 ± 4.4%, compared to the control (56 ± 8.7%, *p* = 0.049). In addition to these changes, there was a pre-G0 peak both at 96 and 120 h that would become significant (*p* < 0.001) in the case of treatment with 500 μM of ZOL.

Regarding the ZOL/BCP association, the percentages of the cells are distributed through the cell cycle similarly to the control.

### 3.8. Types of Cell Death

As can be seen in Figure 8, the number of living cells and the number of apoptotic cells are influenced significantly by the exposure to ZOL. For cells incubated for 96 h with 100 μM of ZOL (ZOL100), there was a significant decrease (*p* < 0.001) in the population of living cells from 82.6 ± 6.6% (control cultures) to 66.5 ± 9.9%. This was accompanied by a significant increase (*p* < 0.001) of the population of cells undergoing apoptosis, from 8.4 ± 3.8% in the control cultures to 20.3 ± 3.7%.

In the case of cells treated with 500 μM of ZOL (ZOL500), there was a marked decrease in the population of living cells to 63.5 ± 4.3% (*p* < 0.001), as well as an increase in apoptosis (16.9 ± 8.4%, *p* = 0.026). On the other hand, in the case of the cells treated with the combination ZOL100/BCP, there were no alterations in the various cell populations compared to the control. When the treatment was performed with a ZOL500/BCP combination, the significant decrease of the viable cell population (*p* < 0.01) was maintained; however, no significant changes in populations in apoptosis and necrosis were seen. The protective effect of BCP was also evidenced by the significantly lower number of viable cells treated with ZOL100 versus those treated with the ZOL100/BCP association (*p* = 0.036).

When the evaluation was carried out after 120 h of incubation, there was a significant decrease in the population of living cells in both the treatment ZOL100, where there was a reduction of 81.2 ± 5.1% to 60.7 ± 4.7% (*p* = 0.011), and in the treatment ZOL500, where the percentage of viable cells decreased to 48.1 ± 5.6% (*p* < 0.001).

## 4. Discussion

The pathophysiology of the osteonecrosis of the jaw (ONJ) is multifactorial. The lack of treatment options emphasizes the medical and scientific inability to define the preponderant causality. BPs inhibit osteogenic cells, osteoblasts, osteocytes, and osteoclasts [33,34,35,36], restrict angiogenesis by inhibiting endothelial cells [37], and reduce the viability of oral keratinocytes, fibroblasts, and epithelial cells, compromising oral mucosal healing [14,38,39,40,41]. These circumstances make soft tissue unable to cover the surgical wound, maintaining the exposure of the bone [16,41,42]. Continuous exposure to oral flora associated with BP-induced bone turnover maintains the inflammatory process involved in ONJ [43,44].

In order to address BRONJ, we hypothesized that calcium phosphate compounds, which are used as bisphosphonate carriers for therapeutic purposes, might have a beneficial role due to their capacity to capture ZOL. To verify this, we selected two calcium phosphate ceramics, TCP and BCP, and evaluated their interaction with ZOL by UV/visible spectroscopy and by elemental analysis. The decrease in ZOL absorbance in the presence of TCP and BCP was corroborated by elemental analysis studies, where organic elements where found in BCP previously stirred with ZOL. Prior X-ray diffraction studies also proved a decrease in ZOL intensity peaks in the presence of hydroxyapatite, not due to significant structural change but due to a stable structural link between ZOL and hydroxyapatite [45]. Freire et al. also supported the structural adsorption of ZOL to hydroxyapatite [46].

In our study, although TCP is presented as the most effective in ZOL adsorption, it has not been ignored that this CP degrades rapidly in vivo [47,48]. Due to its low mechanical strength and rapid reabsorption, TCP might not be able to capture ZOL over a long time period. Therefore, we optimized a chemical reaction in which ZOL was placed in contact with BCP, a CP with superior properties, undergoing adsorption by the latter. 

During healing, the proper reconstruction of soft tissues requires the coordinated interaction of keratinocytes, fibroblasts, endothelial cells, and bone marrow cells, which results in multiplication, proliferation, and migration [49,50]. Fibroblasts have a major role in filling the defect, and they provide the extracellular matrix proteins [51,52].

Therefore, HGF primary cultures were established after the collection of a gingival biopsy through the adaptations of earlier works [20,21,53,54]. These cells, whose morphology was evaluated routinely through the work, were used for subsequent studies. 

Predictably, we observed that ZOL significantly decreased the HGF metabolic activity after 72 h of exposure in a manner dependent of the concentration. Earlier studies also showed ZOL cytotoxicity, not only in HGF [37,55,56,57] but also in oral keratinocytes [58,59], epithelial cells, and other cell types [60].

However, when HGF were exposed to ZOL–BCP, a significant protective effect was observed. The inhibitory effect of ZOL was completely reversed with the association with BCP in the case of exposure for 48, 72, and 96 h for all concentrations of ZOL tested and for 120 h with up to 100 μM of ZOL. Hence, the evaluation of metabolic activity confirmed our hypothesis, showing that ZOL adsorption by calcium phosphate ceramics limits its cytotoxicity.

As migration is a prerequisite for fibroblasts to participate in wound healing, the remodeling of blood vessels, and tissue formation, migration and motility ability were accessed. The scratch assay is the first-line assay to evaluate healing capacity of cells subjected to a stimulus [61]. We verified that untreated HGF fill 60% of the space in 72 h, and confluence is reached after 96 h. As in prior studies [14,37], ZOL-treated cells’ inability to close the space between edges was evident and was accompanied by cellular morphology alterations. ZOL/BCP restored the migration ability, as cultures behaved similarly to controls. 

The confirmatory viability studies reinforced the cytotoxicity of ZOL. Agis and colleagues also verified the cytotoxicity of ZOL, even for gingival fibroblast cultures treated for 72 h [62].

After 120 h, we found less than 25% of viability in ZOL-treated HGF cultures. However, for the same time of incubation ZOL/BCP-treated cell, viability was like the untreated control cell cultures. 

During the G1 phase, cell organelles are formed, and cells grow. In the S phase, self-replicating of DNA molecules occurs and in the G2 phase, and the synthesis of molecules necessary for cell division takes place [63]. We observed a slight increase of cells in the G1 phase after exposure to a high concentration of ZOL (500 μM), but no alterations were found in ZOL/BCP-treated HGF. Prior studies by Manzano-Moreno and colleagues showed that the cell cycle of bisphosphonate-treated osteoblast-like cells was arrested in G1, and the percentage of cells in apoptosis was increased [64].

In order to deepen the mechanisms underlying ZOL cytotoxicity and the ZOL/BCP effect, an evaluation of types of cell death was performed. After 96 h of exposure to ZOL, we found an increase in the percentage of cells in apoptosis that determines a decrease in the viable cells. After 120 h of ZOL exposure, cell death shifts to late apoptosis (100 µM) and necrosis (500 µM). Apoptosis was previously indicated as a consequence of ZOL exposure in HGF [14,55,59,62,65] and in keratinocytes [59,66]. These studies reinforce CP ceramics’ protective effect, as ZOL/BCP-treated cells did not show any significant alterations in the populations in cell death, and the decrease of cell viability was negligible. In the future, it would be interesting to further explore types of cell death and to see if senescence pathways are associated with this process.

Our results validate our hypothesis. BCP interaction with ZOL reduces or even nullifies its toxicity in HGF. This finding represents a potential solution to BRONJ in the case of patients undergoing therapy with ZOL. Some studies with animal models of BRONJ have already been described [67,68,69,70]. In order to verify our hypothesis in vivo, pre-clinical studies in animals are currently being performed by our group.

## 5. Conclusions

ZOL had a strong cytotoxic effect in HGF, observed by the reduction of metabolic activity, migration, and cell viability, as well as by the increase of cells in apoptosis. The ZOL/BCP association reduced or abolished ZOL toxicity, as was demonstrated by the absence of differences related to control. In the future, it would be interesting to further explore types of cell death and to see if senescence pathways are associated with this process. Nevertheless, in vivo studies are already being performed. 

This work confirmed our hypothesis that BCP, a material already used in dentistry, has a protective effect regarding ZOL cytotoxicity. BCP application in the surgical wound might capture ZOL released from the bone, decreasing its bioavailability. This finding might represent a potential solution for BRONJ in the case of patients undergoing therapy with ZOL.

## Figures and Tables

**Figure 1 materials-12-01840-f001:**
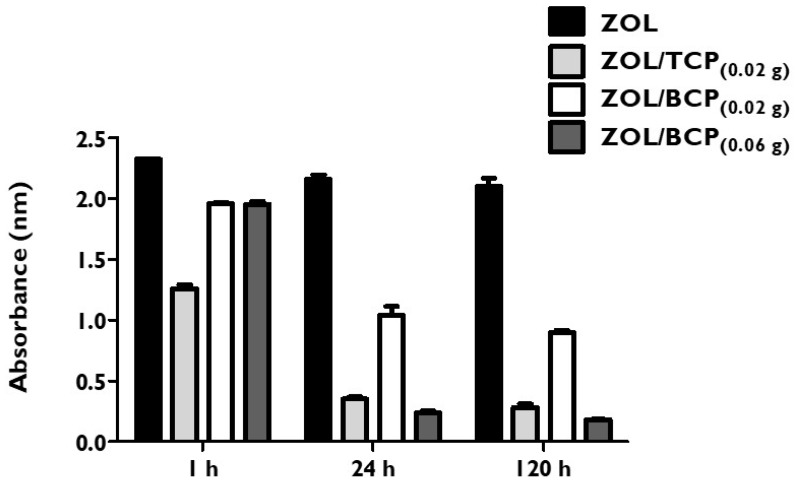
Absorbance (arbitrary units) of the zoledronic acid (ZOL) solution and of the solutions resulting from the mixture with 0.02 g beta-tricalcium phosphate (TCP), 0.02 g biphasic calcium phosphates (BCP), and 0.06 g BCP after stirring for 1 h, 24 h and 120 h.

**Figure 2 materials-12-01840-f002:**
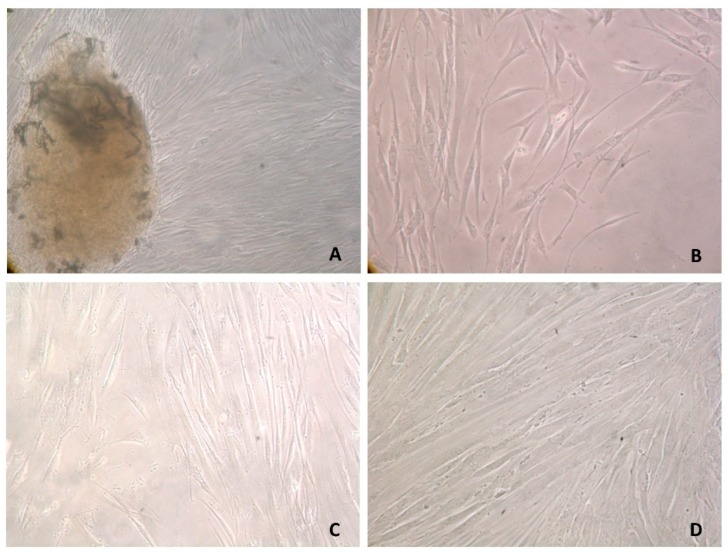
Human gingival fibroblasts (HGF) primary cultures. (**A**) Cells emerging from an explant, 25 days after culture establishment, 100×. (**B**) Cells emerging from an explant, 34 days after culture establishment, 100×. (**C**) HGF, one passage, low confluence, 200×. (**D**) HGF, tenth passage, high confluence, 400×.

**Figure 3 materials-12-01840-f003:**
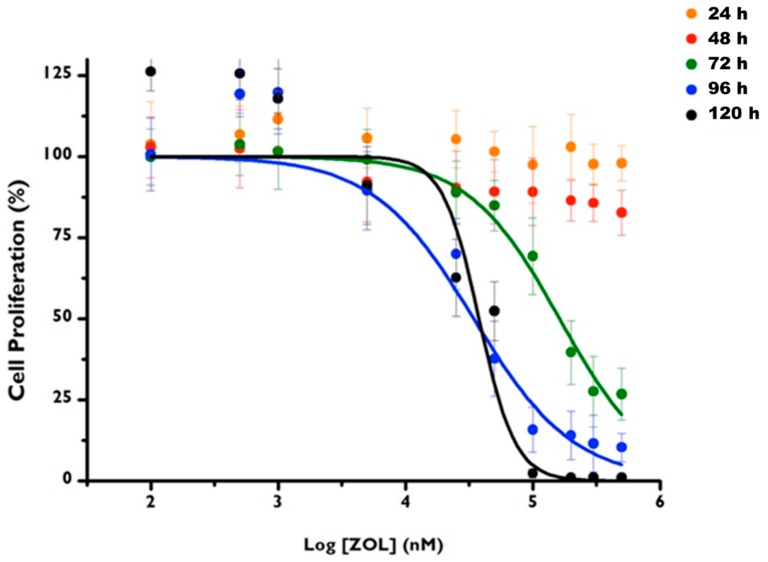
Dose-response curves of HGF primary cell cultures after 24 h, 48 h, 72 h, 96 h and 120 h of incubation with ZOL in concentrations ranging from 0.1 to 1 mM. The bars represent the mean and standard deviation of at least six tests.

**Figure 4 materials-12-01840-f004:**
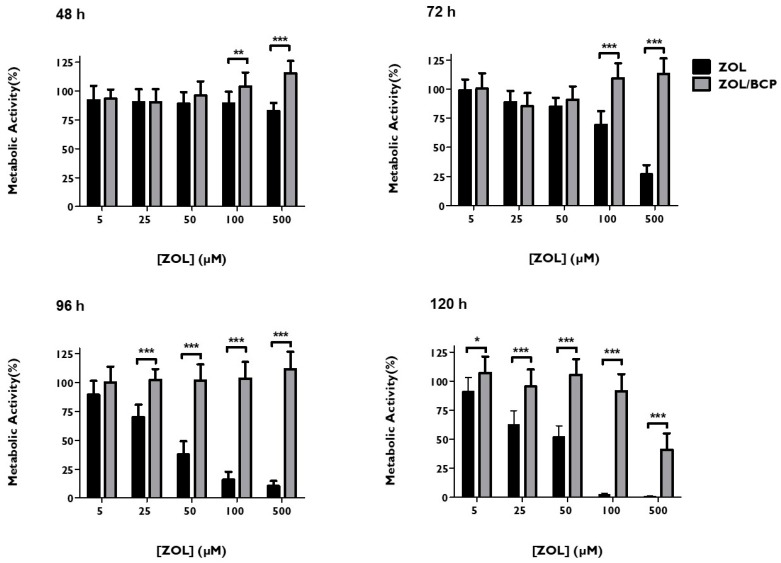
HGF metabolic activity, 48 h, 72 h, 96 h and 120 h after treatment with ZOL and with ZOL/BCP. The results represent the mean and standard deviation of at least six tests. Significant differences are represented with *, where * means *p* < 0.05, ** means *p* < 0.01, and *** means *p* < 0.001.

**Figure 5 materials-12-01840-f005:**
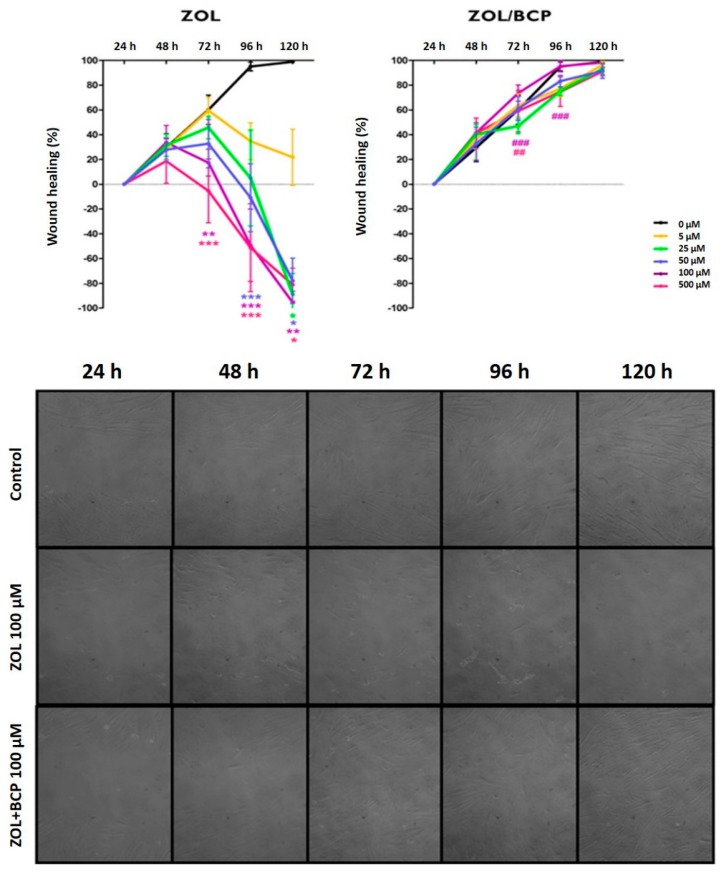
The migration of HGF up to 120 h, comparing cell cultures under treatment with ZOL (left) and with ZOL/BCP (right). The results represent the mean and standard deviation of at least three tests. Significant differences are represented with *, where * means *p* < 0.05, ** means *p* < 0.01, and *** means *p* < 0.001. The images in the bottom of the panel are representative microphotographs of control cell cultures and cell cultures submitted to ZOL and to ZOL/BCP for 24 h, 48 h, 72 h, 96 h and 120 h.

**Figure 6 materials-12-01840-f006:**
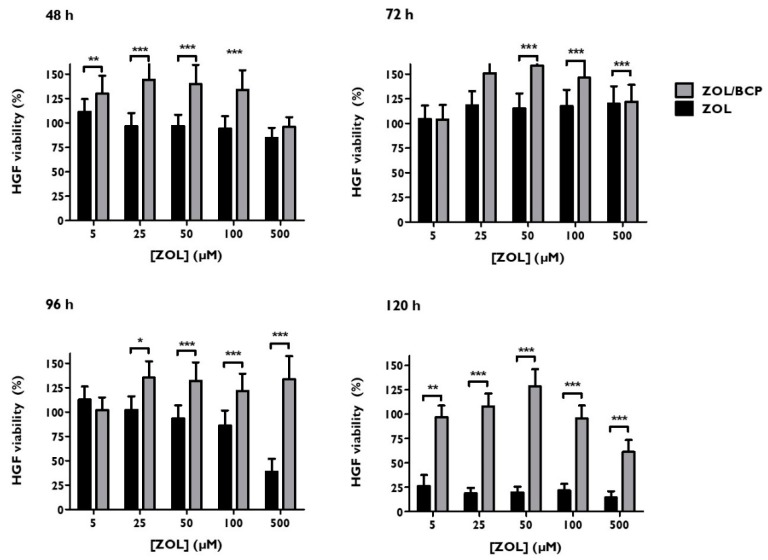
HGF viability, 48 h, 72 h, 96 h and 120 h after treatment with ZOL and with ZOL/BCP. The results represent the mean and standard deviation of at least six tests. Significant differences are represented with *, where * means *p* < 0.05, ** means *p* < 0.01, and *** means *p* < 0.001.

**Figure 7 materials-12-01840-f007:**
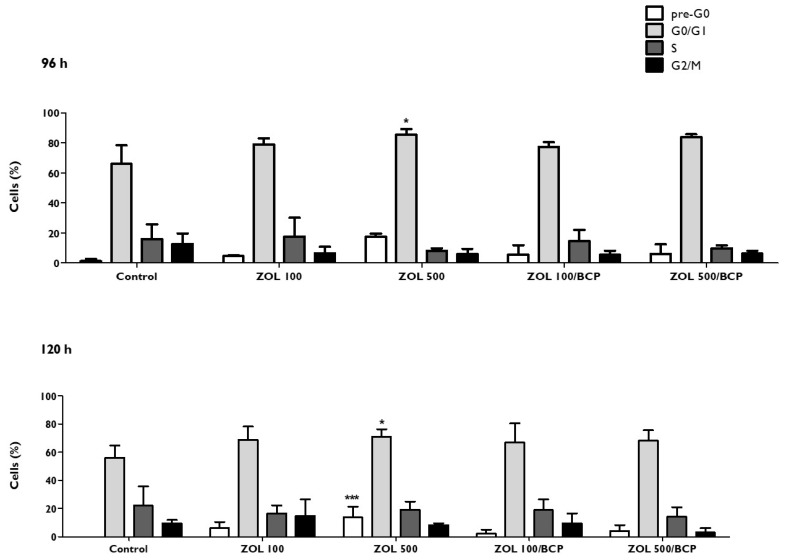
HGF cell cycle, 96 h and 120 h after treatment with ZOL and with ZOL/BCP. The results represent the mean and standard deviation of at least three tests. Significant differences are represented with *, where * means *p* < 0.05, ** means *p* < 0.01, and *** means *p* < 0.001.

**Figure 8 materials-12-01840-f008:**
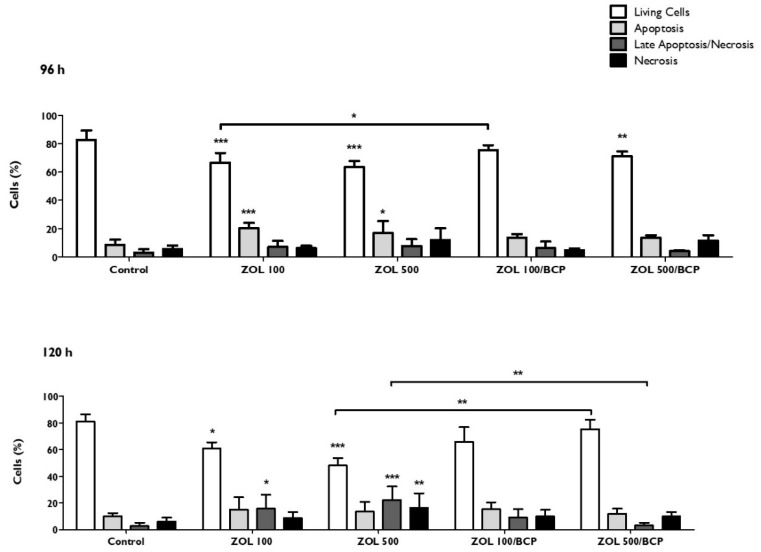
Types of cell death of HGF, 96 h and 120 h after treatment with ZOL and with ZOL/BCP. The results represent the mean and standard deviation of at least four tests. Significant differences are represented with *, where * means *p* < 0.05, ** means *p* < 0.01, and *** means *p* < 0.001.

**Table 1 materials-12-01840-t001:** Median inhibitory concentration of the cultures (IC50) values of human fibroblast primary cultures submitted to ZOL for 72 h, 96 h and 120 h, the respective confidence interval at 95%, and the determination coefficient of the curves.

Time	IC50 (μM)	Confidence Interval 95%	R^2^
72 h	162.1	[129.2; 203.3]	0.997
96 h	34.3	[19.7; 59.9]	0.943
120 h	37.3	[25.1; 56.8]	0.932

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
