# Peer review of "Synthetic Calcium Phosphate Ceramics as a Potential Treatment for Bisphosphonate-Related Osteonecrosis of the Jaw"

_materials, 2019, doi:10.3390/ma12111840_

Round 1
Reviewer 1 Report
Paulo et al., show that calcium phosphate can adsorb zoledronate in HGFs, thereby providing a protective effect to zoledronate toxicity. This can be exploited to help patients treated with nitrogen-containing bisphosphonates who suffer from osteonecrosis. While the manuscript is scientifically sound, it would be interesting to the readers to have more literature discussed to understand the severity of BRONJ.
As in vivo studies are lacking, it would be beneficial to tie the findings in the context of known literature utilising in vivo models.
Author Response
Paulo et al., show that calcium phosphate can adsorb zoledronate in HGFs, thereby providing a protective effect to zoledronate toxicity. This can be exploited to help patients treated with nitrogen-containing bisphosphonates who suffer from osteonecrosis. While the manuscript is scientifically sound, it would be interesting to the readers to have more literature discussed to understand the severity of BRONJ.
R: Further information was added to the introduction section.
As in vivo studies are lacking, it would be beneficial to tie the findings in the context of known literature utilising in vivo models.
R: Further information was added to the discussion section.
Reviewer 2 Report
Ref: materials-515699
Title: Synthetic calcium phosphate ceramics as a potential treatment for
bisphosphonate-related osteonecrosis of the jaw.
The current study is on a topic of relevance and general interest to the readers of the journal. It is well organized and the methods were implemented properly. The results are interesting, presented accurately and contribute to the knowledge of the topic. Maybe a proposal for future studies on this research area should be added in conclusions.
Author Response
1. The current study is on a topic of relevance and general interest to the readers of the journal. It is well organized and the methods were implemented properly. The results are interesting, presented accurately and contribute to the knowledge of the topic. Maybe a proposal for future studies on this research area should be added in conclusions.
R: Our considerations regarding further studies were added to the conclusion section.
Reviewer 3 Report
I read this article with interest, as it represents a potential answer/solution for BRONJ. As I understand, further in vivo tests are currently performed.
My comments/suggestions to improve the quality of this article are as follow:
line 61 Please give full designation when you first use an abbreviation, BRONJ
Please consider introducing a 2.1 for the materials used and than continue with 2.2 ZOL-ceramics adsorption etc as methods.
line 102-103 Please explain why you choose those concentrations.
Please explain how and why did you choose the described protocol for primary cell cultures and ZOL/ZOL-BCP treatment.
line 128 Which subsequently studies described are you referring at?
I presume you used some known protocols for all the assays and analysis, please mention these. There is no information regarding to how you made these choices in your study.
Figure 5. Please provide separate caption for the image of ZOL-BCP/ZOL/Control at 24-120h. It would be better to have a different figure number for this image.
Please use capital letter every time you refer to a Figure in the text.
Author Response
I read this article with interest, as it represents a potential answer/solution for BRONJ. As I understand, further in vivo tests are currently performed.
My comments/suggestions to improve the quality of this article are as follow:
1. line 61 Please give full designation when you first use an abbreviation, BRONJ. R: It was corrected in the manuscript.
2. Please consider introducing a 2.1 for the materials used and then continue with 2.2 ZOL-ceramics adsorption etc as methods.
R: It was corrected in the manuscript.
3. line 102-103 Please explain why you choose those concentrations.
R: There are no studies estimating local concentrations of bisphosphonates in maxillary osteonecrosis lesions, accessible to oral epithelium and fibroblasts following intravenous administration. However, the minimum concentrations in the human plasma, immediately after an infusion of zoledronate, are between 1μM and 10μM and in the bone can be up to 100 times superior (Curr. Pharm. Des., 2010, 16, 2950–60; J. Clin. Pharmacol., 202, 42, 1228–36). It was therefore intended to include a range of concentrations from the lowest estimated value to an overestimated value, to mimic the increase of concentration in the presence of a persistent inflammatory process. Moreover, other authors considered concentrations between 0.1μM and 500μM (Clin. Oral Investig., 2010, 14, 35–41).
4. Please explain how and why did you choose the described protocol for primary cell cultures and ZOL/ZOL-BCP treatment.
R: Primary cell cultures are relevant in vitro models and human gingival fibroblast are important cells concerning the healing process. The obtainment of human fibroblast primary cultures with the explant culture method is commonly described and easy to perform. This methodology proved to be effective and reproducible to culture human gingival fibroblasts, therefore it was adopted. Calcium phosphate materials react in aqueous solution and may undergo dissolution, hydrolysis, precipitation and ionic substitutions. These reactions favor the action of calcium phosphates as drug delivery systems or as substances for bone regeneration (Adv. Drug Deliv. Rev., 2012, 64, 1090–110). Although β-TCP [Ca3 (PO4) 2] is the compound that most effectively adsorve zoledronate chemicaly (ZOL), it degrades more easily in vivo (J. Mater. Sci. Mater. Med., 2003, 14, 195–200; J. Periodontol.,1992, 63, 729–35). Considering the slower kinetics of BCP uptake than β-TCP, two different BCP masses (0.02 g and 0.06 g) were used to react with the zoledronate solution, besides β-TCP.
5. line 128 Which subsequently studies described are you referring at ?
R: The studies subsequently described are cell cytotoxicity, cell migration, viability assay, cell cycle analysis and types of cell death. We clarified in the manuscript.
6. I presume you used some known protocols for all the assays and analysis, please mention these. There is no information regarding to how you made these choices in your study.
R: The protocols used in this study are frequently described in the literature for several applications within the field of biomedical sciences and are currently performed and well stablished in our lab. In order to complement the methodology, appropriate references were added to the manuscript.
7. Figure 5. Please provide separate caption for the image of ZOL-BCP/ZOL/Control at 24-120h. It would be better to have a different figure number for this image.
R: The caption was modified in the manuscript.
8. Please use capital letter every time you refer to a Figure in the text.
R: It was corrected in the manuscript.